





**The Coupling of Carbon, Nitrogen and Sulphur Transformational Processes in River Sediments**
**Based on Correlationship among the Functional Genes**
Mingzhu Zhang[1], Yang Li[1], Qingye Sun[1], Piaoxue Chen[1], Xuhao Wei[1]
[1]School of Resources and Environmental Engineering, Anhui University, Hefei, Anhui Province,
230601, China
*Correspondence to*: Qingye Sun (sunqingye@ahu.edu.cn)
**Abstract:** Microorganisms in sediments play an important role in C-, N- and S-cycles by regulating
forms and contents of these elements. The coupled system or synergistic reaction among three
elemental cycles can effectively alleviate the pollution of C, N, and S in sediments. However,
ecological processes coupling C-, N- and S-cycles in sediments are still poorly understood. In order to
understand the ecological processes mediated by microorganisms living in river sediments, a total of
135 sediment samples were collected from Huaihe River and its branches located in the Northern of
Anhui Province, the abundance of functional marker genes (*mcrA*, *pmoA*, *cmo*, *amoA*, *hzo*, *nirK*, *nirS*,
*nosZ*, *dsrB, aprA*), involving in C-, N- and S-transformation, were determined by *q*PCR. The
correlation among functional genes from 135 river sediment samples was calculated. We supposed
that the correlationship among functional genes could be used as a reference index speculating the
coupled systems of C-N-S in this reasearch, then the distinct coupling relation of C-N-S was revealed,
and probable genetic mechanisms were also expounded based on the hypothesis. The study found that
*amoA*-AOA and *dsrB* possibly played a secondary role, while S-functional gene (*aprA*), C-functional



gene (*mcrA*) and N-functional gene (*hzo*) were the key functional genes that participate in the coupled
processes in the elemental biogeochemical cycle. The results also demonstrated that C, N might have
combined effects on the coupling of carbon, nitrogen and sulphur transformation.
**Keywords:** river sediment, coupled systems, C, N, and S cycles, functional genes
**1 Introduction**
Rivers play a substantial part in elemental biogeochemical processes (Aufdenkampe et al., 2011),
which can regulate the carbon (C), nitrogen (N) and sulphur (S) cycles and act as a good indicator of
environmental changes (Crump et al., 2009;Williamson et al., 2008). However, the nutrient elements
(such as carbon, nitrogen and sulphur) originating from domestic sewage, farm drainage, industrial
effluent, etc. flow into the river, and deposit into the sediments (Cheng et al., 2014;Liu et al.,
2014;Fonti et al., 2015), which lead to the deterioration of river ecosystems.
Studies demonstrated that microorganisms in the artificial environments could couple the
transformation processes of different elements by inter-specific cooperation or coordination of
inter-gene from the same species (Zhi and Ji, 2014). In coupling with methane-nitrogen cycle,
anammox-methanogenesis (Bai et al., 2013), nitrite-driven anaerobic methane oxidation (Ettwig et al.,
2010), aerobic methane oxidation-denitrification (AME-D) (Knittel and Boetius, 2008;Modin et al.,
2008;Modin et al., 2007) and denitrification-methanogenesis (Kodera et al., 2017;Wang et al., 2017)



have been confirmed. For the coupling of S and N cycles, Fdz-Polanco et al. (2001) firstly approved
the sulfate-reducing anaerobic ammonium oxidation (SRAO) process to explain "abnormal" losses of
nitrogen and sulfate. And subsequently several laboratory studies were conducted for purpose of
speculate the pathway of SRAO (Rikmann et al., 2012;Zhang et al., 2009;Schrum et al., 2009). The
occurrence of microaerophilic sulfate and nitrate co-reduction system has been previously reported
(Bowles et al., 2012;Brunet and Garciagil, 1996). For the coupling of C and S cycles, the pathway of
sulfate-dependent anaerobic methane oxidation had been discovered, which was common completed
by anaerobic methanotrophic archaea and sulfate-reducing bacteria (M et al., 2003).
Recently, the coupling cycle between different elements in natural or constructed wetlands, such
as methane oxidation coupled to nitrogen fixation (Larmola et al., 2014), methane oxidation coupled
to ammonium oxidation (Zhu et al., 2010), methane oxidation coupled to denitrification (Zhu et al.,
2016;Long et al., 2016;Long et al., 2017;Luo et al., 2017;Zhang et al., 2018), methane oxidation
coupled to sulfate reduction (Xu et al., 2014;Weber et al., 2017;Emil et al., 2016), etc., received
extensive attention. The coupling cycle between different elements was mainly driven by functional
groups from bacteria and/or archaea living in sediments. The enzymes coded by functional gene(s) in
functional groups catalyze each reaction step in the biogeochemical cycle of elements. At presently,
the functional genes have been regarded as appropriate indicators for the related biogeochemical
processes in the C and N cycles (Petersen et al., 2012;Rocca et al., 2014). The development of





molecular biological technique greatly facilitate the quantitation of functional genes in environmental
samples (Lammel et al., 2015;Petersen et al., 2012). Many studies have used the abundance of
functional groups or functional genes involving in elemental cycle to explore the elemental metabolic
pathways in different ecosystems (Bru et al., 2011;Xie et al., 2014;Smith et al., 2015).
Studies have shown that the microbial functional groups that complete a biogeochemical reaction
may come from different microbial groups, and the same type of bacteria or archaea may also
participate in different steps of the biogeochemical cycle. Therefore, compared with the microbial
functional group, the correlationship among the functional genes can not only better reveal the
coupling relationship of elemental metabolic processes in environmental media (especially for some
natural ecosystems or more complex environmental media, such as sediments), but also predict some
undetected coupling reactions. The main aims of this study were: (1) to analyze the correlation among
the different functional genes related to some known coupled metabolic processes in sediments, and (2)
to predict the possible coupling systems in sediments based on the correlation among the functional
genes; and (3) to illustrate the key functional genes that participate in certain specific metabolic
processes or steps in the elemental biogeochemical cycle.

**2 Materials and Methods**
**2.1 Site description**





The Huaihe River is located in the eastern China, watershed area of approximately 270,000 km$^2$,
involving 5 Provinces (Henan, Anhui, Shandong, Jiangsu and Hubei) and 165 million population,
situated in a transition zone of northern-southern climates in China (Meng et al., 2014;He et al., 2015)
and belongs to monsoon climate from north subtropical to south warm temperature, and from humid
to semihumid-semiarid. The average annual precipitation and the annual evaporation in the basin are
some 883mm and 900-1500mm, respectively. The rainfall of flood season (June to September) usually
amounts to 70% of the annual value. The average annual temperature ranges 13.2-15.7℃ and frost
free period is about 200-240 day. In the basin, a complex interaction of meteorological and
hydrological processes frequently trigger and exacerbate flood and drought events (Wang et al.,
2014;Zhang et al., 2015). Water resources per capita and per unit area in Huaihe River basin is less
than one-fifth of the Chinese average. And more than 50% of the water resources are over-exploited
(Jiang, 2011). In this basin, agricultural cultivation and livestock have a long history. Textile,
household appliances, steel, cement and fertilizer, as the major industries, mainly distribute along the
main stream and branches of Huaihe River, which are running through the main economic areas in the
middle-eastern of China (Tian et al., 2013). In recent decades, a large number of nutrient from farm
drainage, domestic sewage, industrial effluent, etc., had entered into the main stream and branches and
deposited in the river sediment.
**2.2 Sample collection and pretreatment**



In this study, the main stream and the leftward branches located in the Anhui Province were
chosen to do as the investigated area. The length of main stream of Huaihe River in Anhui Province is
more than 400km and its leftward branches in Anhui Province mainly include Honghe river, Guhe
River, Runhe River, Shayinghe River, Xifeihe River, Cihuai River, Qianhe River, Guohe River,
Beifeihe River, Xiehe-Huihe River, Tuohe River, Bianhe River, Suihe River, etc. All branches
investigated are situated in Wanbei plain, which is a part of North China Plain. A total of 135 sections
from main stream and its branches were chosen to collect the sediment samples. Before field sampling,
all of sampling sections were set by the remote sensing map (Fig 1).
In each sampling section, 5 subsamples of surface sediment (depth: 0-10cm) were collected by
Pedersen sampler and then mixed into a sample. The sediment sample was immediately loaded into a
sterile self-sealing bag and then stored in the incubator with 4℃ in the field. After returning to
laboratory, each sample was divided into two parts, one was used to analyze the chemical properties
and another was directly extracted DNA for the molecular biological test. The samples using to
analyze chemical properties were desiccated by the method of vacuum freeze drying and then
screened. After screening, the samples were loaded into the self-sealing bag and then stored in the
refrigerating cabinet with -20℃ until the chemical analysis was carried out.
**2.3 Chemical analysis of sediment samples**
The pH was assessed by the Mettler Toledo FE20 pH meter ($sediment_{mass}$: $H_2O_{volume}$=1g: 5ml).



The organic matter (OM) was determined by the loss of ignition (LOI) in a muffle furnace at $550 \pm 5$ ℃
for 6 h. The total nitrogen (TN) content was measured using the Kjeldahl method. Concentrations of
$NH_4^+$-N, $NO_3^-$-N and $NO_2^-$-N and in sediment samples were determined using a UV-1800
spectrophotometer (Shimadzu, Kyoto, Japan). SMT (standard measurement and test) (Ruban et al.,
2001) method is used to measure the total phosphorus (TP) inorganic phosphorus (IP) and organic
phosphorus(OP) in the sediment.
**2.4 DNA extraction**
Total DNA in sediment samples were extracted by using the PowerSoil® DNA isolation kit (Mo
Bio Carlsbad USA) in accordance with the manufacturer's instructions. Each extracted genomic DNA
was preserved at −20°C until use.
**2.5 Real-time fluorescent quantitative PCR**
Quantitative analyses of functional genes, including *amoA* of AOA, *amoA* of AOB, *hzo*, *nirK*,
*nirS, nosZ*, *mcrA*, *pmoA*, *dsrB* and *aprA*, were performed. The information on the primers selected for
amplification are listed in supporting information (Table S1). Real-time PCRs were implemented on a
Stepone real-time PCR system (Applied Biosystems USA). Each PCR mixture (10 uL) was composed
of 5uL of Bestar® SYBR qPCR Master Mix Ex TaqTM II (2×), 0.25 uL of each primer (concentration
of 10 uM), 0.2 uL of ROX reference dye (50×), 3.3 uL of ddH$_2$O and 1uL of template DNA (Bestar
Biosystem, German). After generating PCR fragments of the respective functional genes using M13



PCR from clones, standard curves for real-time PCR were prepared based on a serial dilution of
known copies of PCR fragments. The $R^2$ value of each standard curve was above 0.99.
**2.6 Data analysis**
To further investigate the interaction among the environmental parameters, pearson correlation
analysis was applied to determine the significant correlations among the chemical properties.
Correlation analysis was calculated to evaluate ecological associations among different functional
marker genes involving in C-, N- and S-transformation using SPSS Statistics 20 (IBM, USA).
Network graph was employed to investigate the key functional genes and nutrient elements of
affecting the coupling transformation of C, N and S.
Stepwise regression models between functional genes and chemical parameters were established
by using SPSS Statistics 20 (IBM, USA). In stepwise regression analysis, environmental parameters,
(i.e. pH, OM, $NH_4^+$-N, $NO_3^-$-N, $NO_2^-$-N, TN, IP, OP and TP) were used as candidate variables to
integrate with functional genes related to C, N and S cycles.

**3 Results**
**3.1 Chemical properties of river sediments**
Table 1 presented the main chemical properties of 135 sediment samples. The pH values of river
sediments were alkaline (with a mean of 7.78) and exhibited a lower coefficient of variance (CV) in



all of chemical properties detected. TN displayed a higher CV among the different sampling sections
rather than OM and TP. In 135 sections investigated, the content of inorganic nitrogen in sediments
displayed a following order: $NH_4^+$-N > $NO_3^-$-N > $NO_2^-$-N, and $NO_3^-$-N contents among different
sections showed the highest CV in inorganic nitrogen. IP content with a lower CV is higher than OP
content in sediments. In five sections (i.e., sections C1, Q2, T3, TA1 and G6) with higher OM,
$NH_4^+$-N, TN and TP, there were three sections (C1, TA1 and G6) locating in the farmland area. The
first branch of the Huaihe River generally exhibited a lower content of nutrients rather than the
secondaty branches, especially OM, $NH_4^+$-N, $NO_3^-$-N and TN contents in sediments. Data analysis
presented that OM (29.50±13.98 g $kg^{-1}$), $NH_4^+$-N (34.92±34.33 mg $kg^{-1}$), $NO_3^-$-N (7.01±6.85 mg $kg^{-1}$)
and TN (0.41±0.34 g $kg^{-1}$) in the sediments of Guohe River (a first branch of the Huaihe River) were
significantly lower than those in the sediments of its secondary branches (OM: 43.54±21.68 g $kg^{-1}$;
$NH_4^+$-N: 73.45±58.09 mg $kg^{-1}$ ; $NO_3^-$-N: 35.35±20.01 mg $kg^{-1}$ and TN: 0.85±0.66 g $kg^{-1}$, $p<0.05$).
The similar characteristics were found in the Shayinghe River (a first branch of Huaihe River) with
the secondary branches.

Data analysis indicated that there was a significantly positive correlation among the different

chemical properties except for the pH and $NO_2^-$-N (Fig 2). The higher positive correlation between
OM and TN in sediments indicated that both had the same source.
**3.2 Quantities of functional genes related to C, N and S cycles in river sediments**

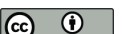



In 13 functional genes investigated in this study, the abundance of *dsrB* and *pmoA*1 genes was
relative higher, and that of *hzo* and *aprA* genes lower (Table 2).
For N-cycling genes, the abundance of *amoA*-AOB was substantially lower as compared to
*amoA*-AOA. Comparing to *nirK* and *nosZ*, *nirS* displayed higher abundance. In the functional genes
related to C-cycle, the *mcrA* abundance exhibited the highest coefficient of variance. Table 2 also
demonstrated that in contrast to *pmoA* and *pmoA*2 genes, type II methanotrophs possessing the *pmoA*1
gene were predominant. In two genes involving in sulfate reduction, the abundance of *dsrB* gene was
significantly more than that of *aprA* gene in sediments.
All of the functional genes investigated in this study displayed higher CV (67.38%-317.86%),
indicating a significant difference in abundance of detected N-, C- and S-cycling genes among 135
river sections.
Table 3 displayed the correlation coefficient among 13 functional genes involving in C-, N- and
S-cycle in sediments. In the functional genes involving in N-cycle, abundances of *cmo*, *hzo*,
*amoA*-AOB, *nirS*, *nosZ* genes were correlated between each other. Meanwhile, abundances of
methanotrophic (*pmoA*, *pmoA*1, *pmoA*2), *mcrA* genes were correlated between each other in C-cycle.
With regard to S-functional genes, no direct relationships between *dsrB* and *aprA* were found.
Concerning the correlations among N-, C- and S-functional genes, the methanotrophic (*pmoA,*
*pmoA*1 and *pmoA*2) genes were correlated with the abundance of *nosZ*. The abundance of *mcrA* gene





had a positive correlation with denitrifying genes (*nirK*, *nirS* and *nosZ*) and *hzo* genes. It was noted
that the *dsrB* and *aprA* gene abundance were positively correlated with *hzo* gene abundance.
Interestingly, positive correlation was also found between the abundance of *aprA* gene and
C-functional genes (*mcrA*, *pmoA*, *pmoA*1and *pmoA*2).
**4 Discussions**
The cycles of carbon, nitrogen and sulphur in environment are made up of a series of chemical
reactions (Parey et al., 2011;Lammel et al., 2015). For the sediment containing a large amount of
organic matter and being in the state of reduction, the oxidation-reduction reaction should be the most
important chemical reaction (Vincent et al., 2017). The substance of the oxidation-reduction reaction
is the gain or loss of electrons or the offset of share electron pair. In river sediment, some elements get
electrons to be reduced, while other elements lose electrons to be oxidized in the oxidation-reduction
reaction. The enzymes from microorganisms, as catalyzer, can accelerate the oxidation-reduction
reactions in sediment (Kandeler et al., 2006;Rocca et al., 2014;Parey et al., 2011). Although sediment
is an important place of elemental cycles, ecological processes regulating methane, nitrogen and sulfur
cycles are poorly understood.
**4.1 Coupling of methane / nitrogen cycles in river sediments**
Bai et al. (2013) revealed that the methanogenesis could coexist with anammox in a single
anaerobic reactor. Based on the hypothesis of this research, there was a positive correlation between



the abundance of *hzo* gene and *mcrA* gene, predicting that methanogenesis and anammox could work
together, which also proved that anammox coupled to methanogenesis (Fig 3).

Studies showed that coupling the nitrate reduction and anaerobic digestion to form a bioreactor,

in which denitrification and methanogenesis process can be carried out simultaneously. The coupled
process could handle the high-strength carbon- and nitrate-containing wastewater, which had received
extensive attention recently (Chen et al., 2009;Sun et al., 2015;Kodera et al., 2017). Based on our
hypothesis, the abundance of *mcrA* gene was positively correlated with denitrifying genes (*nirK*, *nirS*
and *nosZ*) in this study, which can also speculate that simultaneous denitrification and methanogenesis
(SDM) process might occurred (Fig. 3). The simultaneous removal of carbon and nitrogen in the
anaerobic environment through methanogenesis and denitrification was proved to be achievable (Chen
et al., 2009).

$$CH_3OH + NO_3^- \rightarrow N_2 + CO_2 + OH^- + H_2O$$

$$CH_3OH \rightarrow H_2O + CO_2 + CH_4$$

Du et al. (2017) confirmed that it existed in reactor that a novel partial-denitrification combied

with anammox process, since the nitrite for anammox could be acquired from partial-denitrification
process. In our study, the abundance of *hzo* gene showed positive correlations with the denitrifying
genes (*nirK*, *nirS* and *nosZ*), suggesting that denitrification might cooperate with anammox. Bai et al.
(2013) proposed that an integrated process was developed by an anaerobic reactor, in which



methanogenesis, denitrification and anammox were coupled, with methanogenesis first, then

denitrification and anammox simultaneously. Accordingly, the whole abundance of *mcrA* gene was the

highest compared with denitrifying genes (*nirK*, *nirS* and *nosZ*) and *hzo* gene in this study. Therefore,

we postulated the plausible stoichiometric equations, which were deciped in table S2.

Methane oxidation coupled to denitrification consisted of nitrite-driven anaerobic methane

oxidation (Ettwig et al., 2010) and aerobic methane oxidation coupling to denitrification (Zhu et al.,

2016). This research exhibited that methanotrophic (*pmoA, pmoA*1 and *pmoA*2) genes and *cmo* gene

were positively correlated with denitrifying genes (*nirS* and *nosZ*), which inferred the existence of

aerobic methane oxidation coupled to denitrification (AME-D) process and anaerobic

nitrite-dependent methane oxidation process in river sediments as is hypothesized (Fig 3). According

to the speculation of the electron transfer pathway, since aerobic/anaerobic methane oxidation both are

the processes of releasing electrons, while the released electrons are accepted by denitrification

processes ($NO_2^- \rightarrow NO$ and $N_2O \rightarrow N_2$). To date, the aerobic methane oxidation coupled to

denitrification (AME-D) mechanism still remains obscure, and relevant studies have been carried out

to propose different explanations of AME-D progress (Stein and Klotz, 2011); (Modin et al., 2007).

Zhu et al. (2016) summarized the potential energy reactions included in AME-D process. Under

anaerobic conditions, $NO_3^-$ and $NO_2^-$ played a crucial role in supplying electron acceptors in

denitrification processes (Zhu et al., 2016), a tentative inference about AME-D progress on this result



is depicted in table S2. Ettwig et al. (2010) confirmed the existence of nitrite-driven anaerobic
methane oxidation and explained the source of $O_2$ and the production of $N_2$. Dedicated stable isotope
studies showed that this organism could make its own molecular oxygen from nitrite via nitric oxide.
The produced oxygen was mainly used to oxidize methane in an anaerobic environment according to
the expected stoichiometry:

$$3CH_4 + 8NO_2^- + 8H_+ \rightarrow 3CO_2 + 4N_2 + 10H_2O$$

In our study, methanotrophic genes (*pmoA*, *pmoA*1 and *pmoA*2) were positively correlated with
*amoA*-AOB, which can predict the coupled system of aerobic methane oxidation-aerobic ammonia
oxidation based on the correlationship between the functional genes related to C, N cycles (Fig 3).
Some investigators had confirmed that aerobic methanotrophs could oxidize ammonium through
pMMO, since methane monooxygenase (pMMO) and ammonia monooxygenase (AMO) may be
evolutionarily related (Holmes et al., 1995;Klotz and Norton, 1998). The coupled system might be:

$$CH_4 + NH_4 + +O_2 \rightarrow CH_3OH + NO_2^- + H_2O$$

Recent study had confirmed the co-occurrence of nitrite-dependent anaerobic ammonium and
methane oxidation processes in subtropical acidic forest soils (Meng et al., 2016). Anammox and
nitrite-dependent anaerobic methane oxidation (n-damo) which linked the microbial nitrogen and
carbon cycles are two new processes of recent discoveries (Zhu et al., 2010;Meng et al., 2016). In this
research, the abundance of *cmo* gene had a positive correlation with *hzo*, which also predicted the



coupled system of nitrite-dependent anaerobic ammonium and methane oxidation processes on the
basis of our hypothesis (Fig 3).

**4.2 Coupling of nitrogen / sulphur cycles in river sediments**

Sulfate-reducing ammonia oxidation (SRAO) could simultaneously remove ammonium and
sulfate in one anaerobic reactor, and several published works verified this process could occurred both
in laboratory-scale bioreactors or nature (Fdz-Polanco et al., 2001;Rikmann et al., 2012). Our results
found that the abundance of *hzo* gene had a positive correlation with *dsrB* and *aprA* gene, indicating
the occurrence of sulfate-reducing ammonia oxidation (SRAO) process, which further support our
hypothesis (Fig. 4).
The pathway of sulfites reduced to hydrogen sulfide may be: (1) transforming trithionate and
thiosulfate through three consecutive pairs of electron transfer $(3SO_3^{2-} \rightarrow S_3O_6^{2-} \rightarrow S_2O_3^{2-} \rightarrow S^{2-})$.
(2) losing six electrons directly, and not forming above intermediates, which is called the coordinate 6
electron reaction (Parey et al., 2011). In addition, the process of anammox was responsible for
anaerobic nitrogen removal (Rikmann et al., 2012). At present, the transformation of intermediate
involved in anammox still remains ambiguous and it is reported that the intermediate contained
$NH_2OH$, $N_2H_4$ and $HNO_2$, NO and $N_2O$, etc. Up to now, many investigations have been focused on the
feasible metabolic pathway and reaction equations of the synchronously ammonia and sulfate removal.
Sulfate-reducing ammonium oxidation (SRAO) process was first proposed to explain "abnormal"

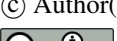



losses of nitrogen and sulfate (Fdz-Polanco et al., 2001).

Possibility of SRAO was noted by Strous et al. (2002), Zhang et al. (2009), Schrum et al. (2009)

$$3SO_4^{2-} + 4NH_4^+ \rightarrow 3HS^- + 4NO_2^- + 4H_2O + 5H^+$$

Coupled with the process of anammox, a summary equation of SRAO was displayed:

$$3SO_4^{2-} + 8NH_4^+ \rightarrow 3HS^- + 4N_2 + 12H_2O + 5H^+$$

In addition to $SO_4^{2-}$, $NO_2^-$ is the most favourable electron acceptor (Rikmann et al., 2012). The

possible half-reactions for SRAO, as suggested by Yang et al. (2009), would be as follows:

$$4NH_4^+ + 8H_2O \rightarrow 4NO_2^- + 32H^+ + 24e^-$$

$$3SO_4^{2-} + 24H^+ + 24e^- \rightarrow 3S^{2-} + 12H_2O$$

Previous research did not clearly indicate the existence of aerobic ammonia oxidation-sulfate

reduction process. In this research, the abundance of *amoA*-AOA gene was positively correlated with
*dsrB* gene, we can speculate the coupled system of aerobic ammonia-sulfate reduction according to
our hypothesis, which might occur through horizontal gene transfer (Fig. 4).

Previous studies had confirmed the existence of microaerophilic sulfate and nitrate co-reduction

system under laboratory conditions (Bowles et al., 2012;Brunet and Garciagil, 1996). The abundance
of denitrifying genes (*nirS*, *nirK* and *nosZ*) had a positive correlation with *aprA* gene, which also
inferred the co-reduction system based on the assumption of this research (Fig 4). Additionally,
several sulfur-reduced compounds ($H_2S$, FeS and $S_2O_3^{2-}$) could act as electron donors for dissimilatory

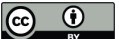



nitrate reduction (Brunet and Garciagil, 1996).

**4.3 Coupling of methane / sulphur cycles in river sediments**

There were two methane-oxidizing mechanisms of aerobic and anaerobic/aerobic oxidation in
sediment. For the coupling of C and S, the pathway of sulfate-dependent anaerobic methane oxidation
had also been discovered (M et al., 2003;Xu et al., 2014). In this study, the positive correlation
between *cmo* gene and *aprA* gene could speculate the coupling relation of anaerobic methane
oxidation-sulfate reduction. Similarly, the abundance of methanotrophic genes (*pmoA*, *pmoA*1 and
*pmoA*2) were positively correlated with *aprA* gene, which can also infer the occurrence of
sulfate-dependent aerobic methane oxidation process, thereby futher supporting the hypothesis (Fig.

5).

The coexistence of methanogenesis and sulfate reduction has been shown before (Maltby et al.,
2018). In this research, the positive correlation between *aprA* gene and *mcrA* gene could also deduce
the presence of methanogenesis within the sulfate reduction zone, which further verified the
hypothesis that the correlationship among functional genes could be used to predict the coupled
systems (Fig 5).

**4.4 Linking the abundance of functional genes and environmental parameters**

In the methane cycle, the *mcrA* gene (methylcoenzyme M reductase) is exclusively linked to
methanogens. Although previous studies have been performed to identify the main factors controlling

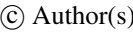



CH$_4$ dynamics from wetlands, the effect of nutrients on CH$_4$ dynamics is poorly understood.
Previously studies found that organic matter, nitrogen and phosphorus was the important regulating
factors in the process of methanogenesis (Yang, 1998). In our study, correlation analysis indicated that
the content of OM, NH$_4^+$-N, NO$_3^-$, TN and OP had significantly positive correlation with the
abundances of methanogenic (*mcrA*) gene (Fig 6). And, the stepwise regression presented a following
regression:   $\log mcrA = 6.359 + 0.006 * \mathrm{NH_4^+} - N + 0.5 * \mathrm{TN} - 0.001 * \mathrm{TP} + 0.325 * \mathrm{pH}$   ($R^2$   =
0.49, *P*<0.001), which indicated that N had a greater effect on *mcrA* than C and P. The abundance of
methanotrophic genes (*pmoA*, *pmoA*1 and *pmoA*2) and *cmo* gene were positively influenced by OM,
NH$_4^+$-N, NO$_3^-$, TN (Fig. 6), suggesting that C and N   co-limitation of the methanotrophs.

In the process of ammonia oxidation, studies indicated that the *amoA*-AOB was generally more

sensitive to higher OM and NH$_4^+$ concentrations (Lammel et al., 2015;Stempfhuber et al., 2014). From
Fig 6, it could be seen that both of OM and NH$_4^+$-N contributed to the increase of the abundance of
AOB and the correlation coefficent between *amoA*-AOB and OM and between *amoA*-AOB and
NH$_4^+$-N was (r=0.424, p<0.01) and (r=0.459, p<0.01), respectively.

The *hzo* gene involving in the anaerobic ammonia oxidation (anammox, NH$_4^+$+NO$_3^-$→H$_2$O+N$_2$)

process (Schmid et al., 2010) mainly mediated by anammox bacteria and was shaped by various
environmental factors in natural habitats (Bai et al., 2015). The abundance of *hzo* gene was mainly
related to the contents of OM, NH$_4^+$, NO$_3^-$, TN in this study (Fig 6).





In this study, all of the denitrifying genes (*nirK*, *nirS* and *nosZ*) was positively correlated with
OM, $NH_4^+$-N, $NO_3^-$-N, TN and OP (Fig 6), which implied that the lower content of nitrogen in
sediments was disadvantageous for denitrification in river sediments.
The *aprA* gene and *dsrB* gene could serve as marker genes for sulfate reduction energy
metabolism (Bae et al., 2015;Meyer and Kuever, 2007). We found that the abundance of *aprA* gene
was positively correlated with OM, $NH_4^+$-N, $NO_3^-$-N, TN, and OP, but no direct correlations
between the *dsrB* copy numbers and any nutrient characteristics of the Huaihe river sediment were
detected. This result is different from study of (Bae et al., 2015), who presented that there was a
positive correlation between *dsrB* gene and TP concentrations.
Integrating the gene abundance data with environmental parameters provided a comprehensive
overview of these interactions related to nitrogen, methane and sulphur cycle, which showed that
among the nutrient characteristics of Huaihe River sediment, organic matter and nitrogen nutrients had
comprehensive and complicate impact on the coupling transformational processes of C, N and S in
river sediment (Fig 6).
Network graph also showed that *amoA*-AOA and *dsrB* played a secondary role in the coupling
transformation of C, N and S, while *aprA, mcrA and hzo* closely participate in the coupling processes
(Fig 6). There was a positive correlation between the abundance of *dsrB* gene and *amoA*-AOA gene,
but *dsrB* gene was not related to *amoA*-AOB gene. It indicated that *amoA*-AOA gene had an important



effect on the coupling process of ammonia oxidation and sulfite reduction. Similarly, in ammonia

oxidation genes (*amoA*-AOA and *amoA*-AOB), *aprA* gene only had a positive correlation with.

*amoA*-AOB gene, which suggested that *amoA*-AOB gene played a key role in the coupling process of

ammonia oxidation and sulfate reduction. Network graph displayed that *aprA* gene played a more

important role than *dsrB* gene in the coupling of N-S and C-S, indicating that the process of sulfite

reduction might occur toughly.

In addition, network graph showed that the *nirS* gene exhibited a greater weight than the *nirK*

gene, indicating that *nirS*-encoding bacteria may take precedence over *nirK*-encoding bacteria in river

sediments investigated in the coupling processes of N-C and N-S. Enwall et al. (2010) held that

different habitat and nutrient content resulted in the differences in abundance of the *nirS*- and

*nirK*-type denitrifiers. Kim et al. (2011) also suggested that both types of denitrifiers apparently

occupy different ecological niches.

**5 Conclusions**

Appropriate marker genes abundance can determine quantification of microbial functional groups.

A direct relationship was established between the nutritional status and the distributions of functional

genes. The C-N, C-S and N-S coupled systems might be inferred in this research based on the

correlationship among functional genes. Compared with other genes, the *amoA*-AOA and *dsrB* played





a minor role in the coupling transformation of C, N and S, while S-functional gene (*aprA*),
C-functional gene (*mcrA*), N-functional gene (*hzo*) were the key functional genes that participate in
the coupled processes in the elemental biogeochemical cycle. Despite the fact that this hypothesis still
has to be verified experimentally it is safe to conclude that C and N might play an important
modulating role in the coupling of carbon, nitrogen and sulphur. Transcription and protein group can
be carried out to further verify if the processes exactly occurred.

**Author contributions**
MZZ, YL, and QYS proposed and organized the overall project. MZZ performed the majority of
the experiments. PXC and XHW gave assistance in sampling and the analyses of chemical properties.
MZZ and QYS wrote the main manuscript text. YL contributed insightful discussions. All authors
reviewed the manuscript.

**Funding**
Financial supports from the National Science and Technology Major Project
(2012ZX07204-004).

**Compliance with ethical standards**





The work has not been published previously and not under consideration for publication
elsewhere. This article does not contain any studies with human participants or animals performed by
any of the authors.

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



**Table 1.** The chemical properties of sediment samples

| Indices | pH | OM g kg$^{-1}$ | NH$_4^+$-N mg kg$^{-1}$ | NO$_3^-$-N mg kg$^{-1}$ | NO$_2^-$-N mg kg$^{-1}$ | TN g kg$^{-1}$ | IP mg kg$^{-1}$ | OP mg kg$^{-1}$ | TP mg kg$^{-1}$ | C/N ratio |
|---|---|---|---|---|---|---|---|---|---|---|
| Mean | 7.78 | 38.45 | 62.40 | 22.04 | 0.24 | 0.87 | 470.93 | 85.82 | 674.52 | 79.12 |
| Median | 7.80 | 34.66 | 44.21 | 12.62 | 0.15 | 0.69 | 448.72 | 73.94 | 644.56 | 52.17 |
| Minimum | 6.08 | 10.31 | 2.87 | 0.10 | 0.01 | 0.01 | 92.93 | 2.16 | 152.65 | 21.17 |
| Maximum | 8.83 | 173.09 | 304.46 | 157.48 | 1.40 | 4.77 | 1631.96 | 509.17 | 2108.46 | 1184.45 |
| CV(%) $^*$ | 5.44 | 56.75 | 86.91 | 124.30 | 94.93 | 85.17 | 39.96 | 65.88 | 39.05 | 145.03 |

Notes: CV—coefficient of variance.

**Table 2.** The abundance of functional genes (copies · g$^{-1}$ dw soil) related to C, N, S cycles

| Functional genes | Mean | CV% | Minimum | Maximum |
|---|---|---|---|---|
| *nirK* | $1.27 \times 10^8$ | 128.94 | $2.17 \times 10^6$ | $9.00 \times 10^8$ |
| *nirS* | $1.55 \times 10^9$ | 163.00 | $6.63 \times 10^6$ | $1.54 \times 10^{10}$ |
| *nosZ* | $1.44 \times 10^8$ | 193.50 | $3.30 \times 10^5$ | $1.73 \times 10^9$ |
| *hzo* | $1.28 \times 10^6$ | 126.67 | $3.33 \times 10^4$ | $1.13 \times 10^7$ |
| *amoA*-AOA | $7.76 \times 10^7$ | 317.86 | $1.16 \times 10^6$ | $2.43 \times 10^9$ |
| *amoA*-AOB | $1.25 \times 10^7$ | 67.38 | $2.32 \times 10^6$ | $6.5 \times 10^7$ |
| *mcrA* | $7.76 \times 10^7$ | 315.34 | $4.31 \times 10^7$ | $2.15 \times 10^{11}$ |
| *pmoA* | $1.32 \times 10^9$ | 248.39 | $3.38 \times 10^6$ | $2.58 \times 10^{10}$ |
| *pmoA*1 | $1.82 \times 10^{10}$ | 210.38 | $9.88 \times 10^6$ | $2.08 \times 10^{11}$ |
| *pmoA*2 | $5.06 \times 10^8$ | 225.95 | $5.29 \times 10^6$ | $6.18 \times 10^9$ |
| *cmo* | $1.18 \times 10^8$ | 91.29 | $4.15 \times 10^6$ | $7.45 \times 10^8$ |
| *dsrB* | $7.82 \times 10^9$ | 146.30 | $1.92 \times 10^8$ | $5.80 \times 10^{10}$ |
| *aprA* | $6.62 \times 10^6$ | 205.35 | $8.62 \times 10^3$ | $1.09 \times 10^8$ |

Notes: CV—coefficient of variance. Denitrification, including *nirS* and *nirK* for nitrite reductase, and *nosZ* for

nitrous oxide reductase; Anammox, including *hzo* for hydrazine oxidoreductase; Nitrification, including *amoA*

encoding bacterial and archaeal ammonia monooxygenase; Methanogenesis, including *mcrA* for the methyl

coenzyme M reductase; Aerobic methane oxidation, including *pmoA* encoding the alpha-subunit of pMMO, in which

*pmoA* gene from conventional type I methanotrophs, conventional type II methanotrophs and type II methanotrophs





possessing the *pmoA*2 gene. Anaerobic nitrite-dependent methane oxidation, including *cmo* gene for M. oxyfera
specific primers; Sulfur reduction, including *dsrB* for dissimilatory sulfite reductase and *aprA* for
adenosine-5'-phosphosulfate (APS) reductase.
**Table 3.** The correlation coefficent among the abundance of 13 functional genes (n=135)

| Items | *hzo* | *cmo* | *AOA* | *AOB* | *nirK* | *nirS* | *nosZ* | *mcrA* | *pmoA* | *pmoA*1 | *pmoA2* | *dsrB* |
|---|---|---|---|---|---|---|---|---|---|---|---|---|
| *cmo* | 0.763** | | | | | | | | | | | |
| *AOA* | 0.042 | -0.04 | | | | | | | | | | |
| *AOB* | 0.492** | 0.575** | 0.361** | | | | | | | | | |
| *nirK* | 0.294** | 0.462** | -0.161 | 0.159 | | | | | | | | |
| *nirS* | 0.366** | 0.617** | -0.188* | 0.253** | 0.810** | | | | | | | |
| *nosZ* | 0.251** | 0.534** | -0.069 | 0.394** | 0.483** | 0.550** | | | | | | |
| *mcrA* | 0.515** | 0.677** | 0.210* | 0.501** | 0.259** | 0.357** | 0.444** | | | | | |
| *pmoA* | 0.503** | 0.510** | 0.142 | 0.308** | 0.135 | 0.260** | 0.316** | 0.594** | | | | |
| *pmoA*1 | 0.566** | 0.788** | -0.107 | 0.503** | 0.414** | 0.586** | 0.540** | 0.481** | 0.402** | | | |
| *pmoA*2 | 0.565** | 0.766** | -0.138 | 0.373** | 0.429** | 0.599** | 0.476** | 0.525** | 0.457** | 0.874** | | |
| *dsrB* | 0.247** | 0.021 | 0.294** | 0.151 | -0.088 | -0.121 | -0.14 | 0.123 | 0.102 | -0.078 | -0.051 | |
| *aprA* | 0.324** | 0.497** | -0.005 | 0.334** | 0.373** | 0.440** | 0.342** | 0.323** | 0.246** | 0.450** | 0.408** | -0.103 |






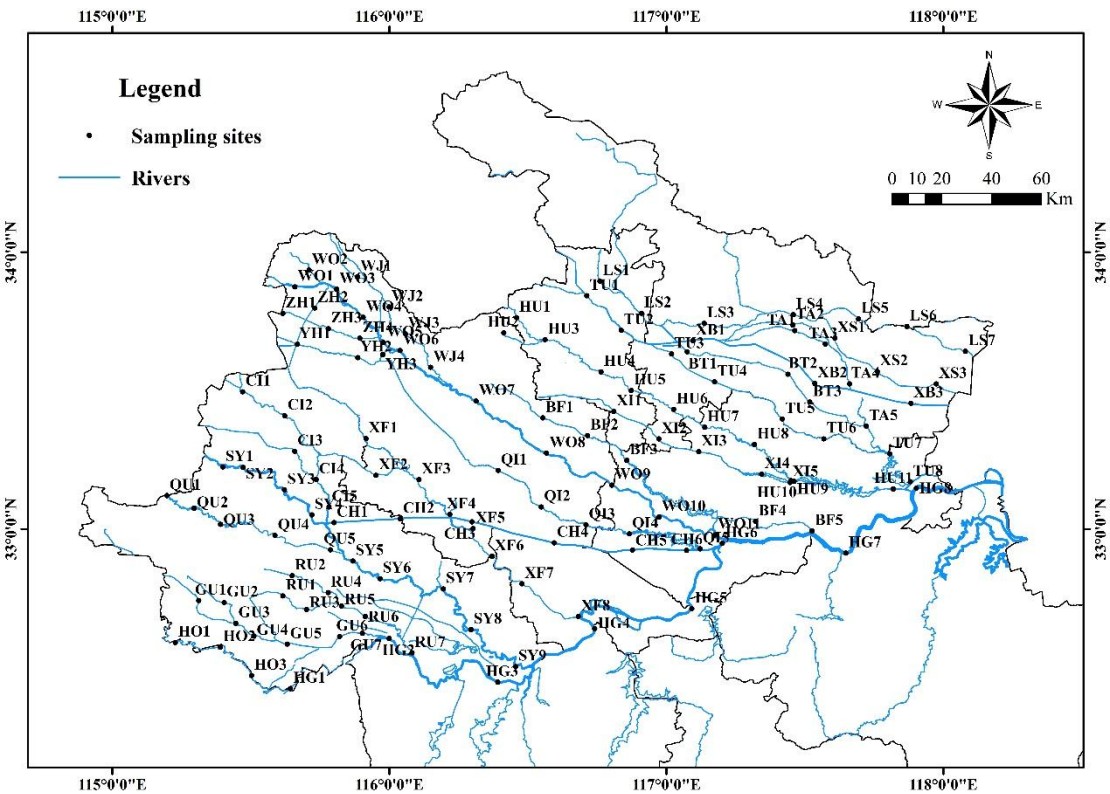

**Fig.1.** Sketch map of sampling sites of rivers in northern Anhui province

Z-Zhaohe River, Y-Youhe River, XS-Xinhe River, X-Xiehe River, XF-Xifeihe River, WJ-Wujiahe

River, W-Guohe River, T-Tuohe River, TA-Tanghe River, S-Shayinghe River, R-Runhe River,

Q-Quanhe River, QI-Qianhe River, L-Suihe River, H-Huihe River, HG-Huaigan river, HO-Honghe

River, G-Guhe River, CH-Cihuai River, C-Cihe River, XB-Bianhe River, BT-Beituohe River,

BF-Beihe River.




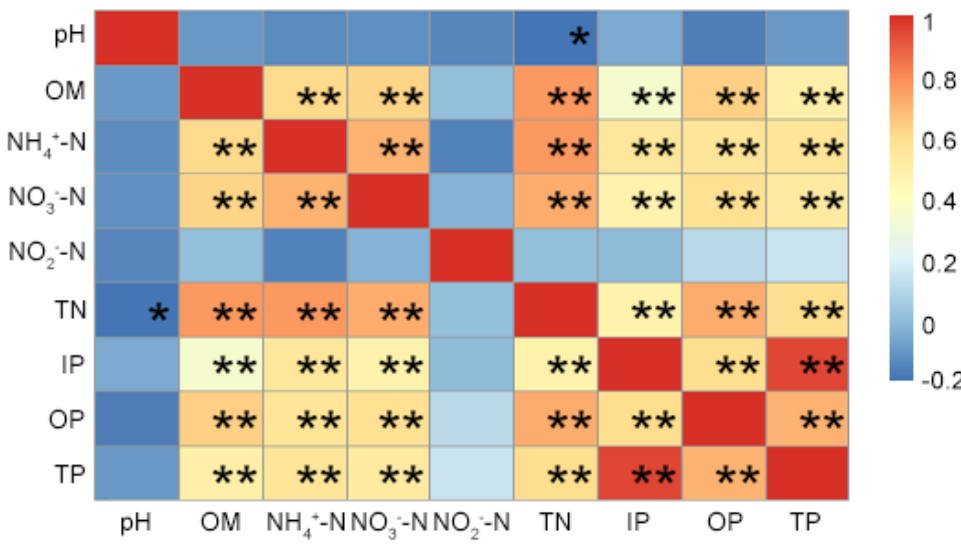


**Fig.2.** The correlation analysis among different chemical properties

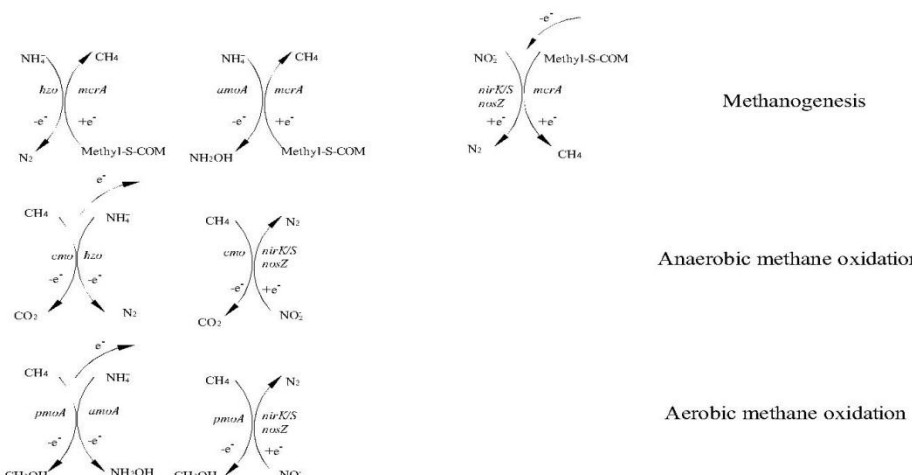

**Fig.3.** Coupling of methane / nitrogen cycles in river sediments





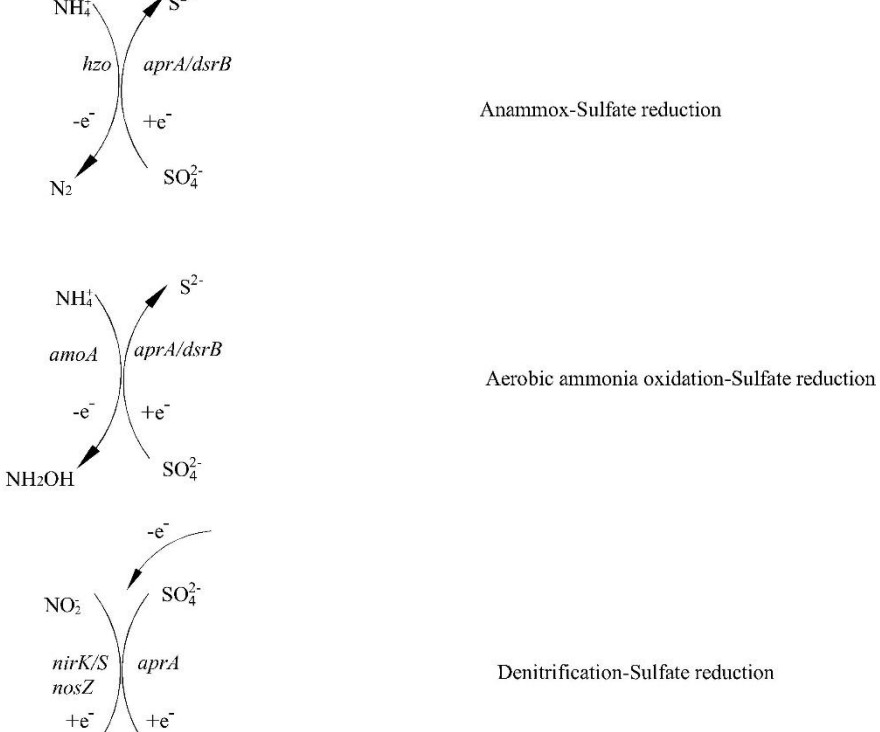


**Fig.4.** Coupling of nitrogen / sulphur cycles in river sediments



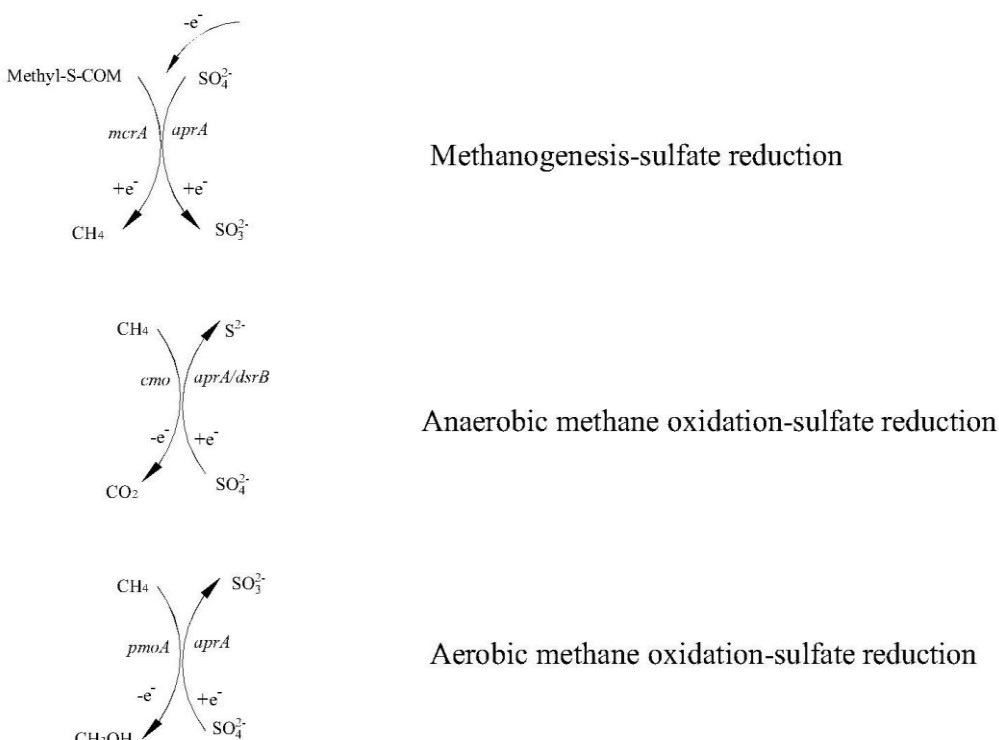


**Fig.5.** Coupling of methane / sulphur cycles in river sediments



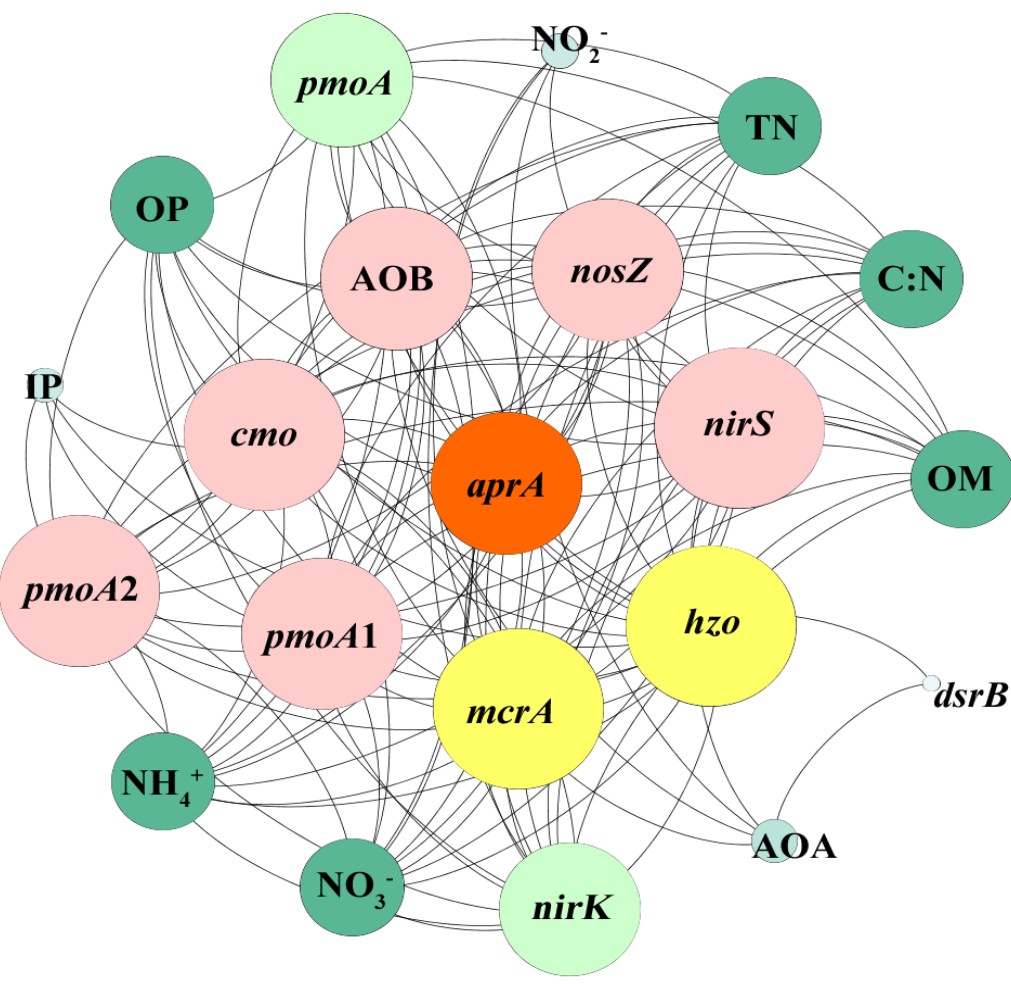


**Fig.6.** Relationships between different chemical properties and functional genes

