# Peer review of "The Coupling of Carbon, Nitrogen and Sulphur Transformational Processes in River Sediments"

_Biogeosciences, 2018_

## Referee Comment (RC1) · Anonymous Referee #1 · 25 Oct 2018

Zhang et al have investigated different functional genes involved in the carbon, nitrogen and sulfur cycle in the river sediments. The study was well designed and the sampling was performed in a large area. The results seem interesting, however, the discussion part should be improved, especially the formulas included in the discussion parts are all from literature, but it has been shown as results in Figure 4. One should be very careful to make conclusions on the reactions without experimental evidence.

There are some specific comments: 1. Page 6 line 104, 'another' should be 'the other'; 2. Page 7 line 112, 'and in sediment' should be 'in sediment' 3. Page 8 line 131,

'interaction' should be 'interactions' 4. The method used for network analysis and other statistical analysis should be described clearly. 5. Page 8 line 144, 'The pH values' should be 'The pH' 6. Page 9 line 147, '135 sections investigated' should be '135 investigated sections' 7. Page 10 line 177, 'correlated between each other' should be 'correlated with each other' 8. Page 18 line 301, 'previously studies' should be 'previous studies' 9. Page 21 line 357-358, please rewrite the sentence. Simply mention C and N does not make sense, especially in the conclusion part.
* * *

---

## Author Comment (AC1) · 20 Dec 2018

Dear reviewer: I am very grateful to your comments for the manuscript entitled "bg-2018-369". Those comments are all valuable and very helpful for revising and improving our paper. Based on your comment and request, we amended the relevant part in manuscript. Some of your questions were answered below. Here, a revised manuscript with the correction sections red marked was attached as the supplemental material and for easy check purpose. The main corrections in the paper and the responds to your comments are as flowing:

[Figure]

Responds to the reviewer's comments:

Comment: Zhang et al have investigated different functional genes involved in the carbon, nitrogen and sulfur cycle in the river sediments. The study was well designed and the sampling was performed in a large area. The results seem interesting, however, the discussion part should be improved, especially the formulas included in the discussion parts are all from literature, but it has been shown as results in Figure 4. One should be very careful to make conclusions on the reactions without experimental evidence. Response: Thank you for your suggestion, all your suggestions are very important, they have important guiding significance for my thesis writing and research work. Necessary change in the statements has been made in the revised manuscript as well as in the referred formulas and figure accordingly. So far, most of studies about coupling of C, N and S transformational processes was conducted in the laboratory (Bowles et al., 2012;MyrtoTsiknia et al., 2015) or wastewater treatment systems (Chen et al. 2009, Wang et al. 2010). The river ecosystem not only contains massive C, N and S, but lives a large number of microorganisms also (Smith et al., 2015;Throbäck et al., 2004;Lin et al., 2016). Microorganisms in sediments play an important role in C-, N- and S-cycles by regulating forms and contents of these elements. And the C, N and S cycles mediated by microorganisms were continuously carried on in river system. In this study, we hypothesize that a natural river ecosystem is a gigantic water treatment system, and the relevance of microbial marker genes abundance in sediment with a similar characteristics to sewage sludge in water treatment system is an appropriate indicator of coupling C, N and S transformational processes in river sediment. It is worth noting that the quantified functional genes have been regarded as appropriate indicators for the related biogeochemical processes in the C and N cycles (Petersen et al., 2012;Rocca et al., 2014). Many studies have used the abundance of functional genes involving in elemental cycle to infer the underlying metabolic pathways in different ecosystems (Bru et al., 2011;Xie et al., 2014;Smith et al., 2015). So this article just proposed a preliminary assumption based on the correlationship among functional genes, then the distinct possible coupling relation of C-N-S was revealed in the river sediments. Thank you for your suggestion. It is very important. Due to your suggestion, I have found some shortcomings in my current work. I will improve my research level and achieve more results according to your suggestions in future work.

References Bowles, M. W., Nigro, L. M., Teske, A. P., and Joye, S. B.: Denitrification and environmental factors influencing nitrate removal in Guaymas Basin hydrothermally altered sediments, Frontiers in Microbiology, 3, 377, 2012. Bru, D., Ramette, A., Saby, N. P. A., Dequiedt, S., Ranjard, L., Jolivet, C., Arrouays, D., and Philippot, L.: Determinants of the distribution of nitrogen-cycling microbial communities at the landscape scale, ISME J, 5, 532-542, http://www.nature.com/ismej/journal/v5/n3/suppinfo/ismej2010130s1.html, 2011. Lin, Z., Sun, X., Peckmann, J., Lu, Y., Xu, L., Strauss, H., Zhou, H., Gong, J., Lu, H., and Teichert, B. M. A.: How sulfate-driven anaerobic oxidation of methane affects the sulfur isotopic composition of pyrite: A SIMS study from the South China Sea, Chemical Geology, 440, 26-41, https://doi.org/10.1016/j.chemgeo.2016.07.007, 2016. MyrtoTsiknia, Paranychianakis, N. V., Varouchakis, E. A., and Nikolaidis, N. P.: Environmental drivers of the distribution of nitrogen functional genes at a watershed scale, FEMS Microbiology Ecology, 91, 2015. Petersen, D. G., Blazewicz, S. J., Firestone, M., Herman, D. J., Turetsky, M., and Waldrop, M.: Abundance of microbial genes associated with nitrogen cycling as indices of biogeochemical process rates across a vegetation gradient in Alaska, Environmental Microbiology, 14, 993-1008, 2012. Rocca, J. D., Hall, E. K., Lennon, J. T., Evans, S. E., Waldrop, M. P., Cotner, J. B., Nemergut, D. R., Graham, E. B., and Wallenstein, M. D.: Relationships between protein-encoding gene abundance and corresponding process are commonly assumed yet rarely observed, Isme Journal, 9, 1693, 2014. Smith, J. M., Mosier, A. C., and Francis, C. A.: Spatiotemporal Relationships Between the Abundance, Distribution, and Potential Activities of Ammonia-Oxidizing and Denitrifying Microorganisms in Intertidal Sediments, Microbial Ecology, 69, 13-24, 2015. Throbäck, I. N., Enwall, K., Jarvis, A., and Hallin, S.: Reassessing PCR primers targeting nirS, nirK and nosZ genes for community surveys of denitrifying bacteria with DGGE, FEMS Microbiology Ecology, 49, 401-417, 2004. Xie,

Z., Roux, X. L., Wang, C., Gu, Z., An, M., Nan, H., Chen, B., Li, F., Liu, Y., and Du, G.: Identifying response groups of soil nitrifiers and denitrifiers to grazing and associated soil environmental drivers in Tibetan alpine meadows, Soil Biology & Biochemistry, 77, 89-99, 2014.

Comment: Page 6 line 104, 'another' should be 'the other'; Page 7 line 112, 'and in sediment' should be 'in sediment'. Page 8 line 131,'interaction' should be 'interactions' Response: Thanks for the referee's kind advice. We are very sorry for our negligence. It has been modified.

Comment: The method used for network analysis and other statistical analysis should be described clearly. Response: Thanks for the referee's suggestion. We have clearly described the method according to the reviewer's suggestions. Spearman analysis was employed to investigate the key functional genes and nutrient elements of affecting the coupling transformation of C, N and S, the p-values in the correlation were adjusted statistically significant (PFDR<0.05). Network analysis was carried out by Gephi software according to the relationships between sediment parameters and functional genes..C, N and S cycles and coupled pathways were carried out following Auto CAD software.

Comment: Page 8 line 144, 'The pH values' should be 'The pH'. Page 9 line 147, '135 sections investigated' should be '135 investigated sections'. Page 10 line 177, 'correlated between each other' should be 'correlated with each other'. Page 18 line 301, 'previously studies' should be 'previous studies'. Response: We are very sorry for our incorrect writing. We have made correction according to the reviewer's comments.

Comment: Page 21 line 357-358, please rewrite the sentence. Simply mention C and N does not make sense, especially in the conclusion part. Response: Thanks for the referee's good evaluation and kind suggestion. It has been modified. The rewrited sentence has been added into the revised manuscript.

Thank you again for your suggestion, I hope to learn more from you. We acknowledge the reviewer's comments and suggestions very much, which are valuable in improving the quality of our manuscript. We tried our best to improve the manuscript and hope that the correction will meet with approval.

Please also note the supplement to this comment:
https://www.biogeosciences-discuss.net/bg-2018-369/bg-2018-369-AC1-supplement.pdf

**Supplement:**

[revised manuscript text omitted]
, the p-values in the correlation were adjusted statistically significant (PFDR<0.05). Network analysis was carried out by Gephi software according to the relationships between sediment parameters and functional genes..C, N and S cycles and coupled pathways were carried out following Auto CAD software.

Stepwise regression models between functional genes and chemical parameters were established by using SPSS Statistics 20 (IBM, USA). In stepwise regression analysis, environmental parameters, (i.e. pH, OM, $NH_4^+$-N, $NO_3^-$-N, $NO_2^-$-N, TN, IP, OP and TP) were used as candidate variables to integrate with functional genes related to C, N and S cycles.

**3 Results**

**3.1 Chemical properties of river sediments**

Table 1 presented the main chemical properties of 135 sediment samples. The pH  of river sediments were alkaline (with a mean of 7.78) and exhibited a lower coefficient of variance (CV) in all of chemical properties detected. TN displayed a higher CV among the different sampling sections rather than OM and TP. In 135 investigated sections , the content of inorganic nitrogen in sediments displayed a following order: $NH_4^+$-N > $NO_3^-$-N > $NO_2^-$-N, and $NO_3^-$-N contents among different sections showed the highest CV in inorganic nitrogen. IP content with a lower CV is higher than OP content in sediments. In five sections (i.e., sections C1, Q2, T3, TA1 and G6) with higher OM, $NH_4^+$-N, TN and TP, there were three sections (C1, TA1 and G6) locating in the farmland area. The first branch of the Huaihe River generally exhibited a lower content of nutrients rather than the secondaty branches, especially OM, $NH_4^+$-N, $NO_3^-$-N and TN contents in sediments. Data analysis presented that OM ($29.50 \pm 13.98$ g·kg$^{-1}$), $NH_4^+$-N ($34.92 \pm 34.33$ mg·kg$^{-1}$), $NO_3^-$-N ($7.01 \pm 6.85$ mg·kg$^{-1}$) and TN ($0.41 \pm 0.34$ g·kg$^{-1}$) in the sediments of Guohe River (a first branch of the Huaihe River) were significantly lower than those in the sediments of its secondary branches (OM: $43.54 \pm 21.68$ g·kg$^{-1}$; $NH_4^+$-N: $73.45 \pm 58.09$ mg·kg$^{-1}$ ; $NO_3^-$-N: $35.35 \pm 20.01$ mg·kg$^{-1}$ and TN: $0.85 \pm 0.66$ 
[revised manuscript text omitted]